

# Gamma-rays induced mutations increase soybean oil and protein contents

Geehan Mohsen[1], Said S. Soliman[1], Elsayed I. Mahgoub[1], Tarik A. Ismail[1], Elsayed Mansour[2], Khairiah M. Alwutayd[3], Fatmah A. Safhi[3], Diaa Abd El-Moneim[4], Rahma Alshamrani[5], Osama O. Atallah[6], Wael F. Shehata[7,8] and Abdallah A. Hassanin[1]

[1] Genetics Department, Faculty of Agriculture, Zagazig University, Zagazig, Egypt
[2] Department of Crop Science, Faculty of Agriculture, Zagazig University, Zagazig, Egypt
[3] Department of Biology, College of Science, Princess Nourah bint Abdulrahman University, Riyadh, Saudi Arabia
[4] Department of Plant Production, (Genetic Branch), Faculty of Environmental and Agricultural Sciences, Arish University, El-Arish, Egypt
[5] Biology Department, Faculty of Science, King Abdulaziz University, Jeddah, Saudi Arabia
[6] Department of Plant Pathology & Microbiology, Faculty of Agriculture & Life Sciences, Texas A&M University, College Station, TX, USA
[7] Department of Agricultural Biotechnology, College of Agriculture and Food Sciences, King Faisal University, Al-Ahsa, Saudi Arabia
[8] Plant Production Department of, College of Environmental Agricultural Science, Arish University, North Sinai, Egypt

Corresponding authors
Diaa Abd El-Moneim,
dabdelmoniem@aru.edu.eg
Wael F. Shehata, wshehata@kfu.edu.sa

## ABSTRACT

Mutation breeding is one of the effective techniques used for improving desired traits such as yield quality and quantity in economic crops. The present study aims to develop oil and protein contents in addition to high yield attributes in soybean using gamma rays as a mutagen. Seeds of the soybean genotypes Giza 21, Giza 22, Giza 82, Giza 83 and 117 were treated with gamma rays doses 50, 100, 200 and 300 Gy. Plants were then scored based on morphological parameters correlated with yield quantity including plant height, seed weight and valuable protein and oil contents. Mutant lines exhibiting the highest yield attributes were selected and used as parents for M2 generation. The M2 progeny was further assessed based on their ability to maintain their yield attributes. Twenty mutant lines were selected and used as M3 lines. The yield parameters inferred a positive effect of gamma irradiation on the collected M3 mutant lines compared to their parental genotypes. 100 Gy of gamma rays gave the highest effect on the number of pods, branches and seeds per plant in addition to protein content, while 200 Gy was more effective in increasing plant height, number of pods per plant, and oil content. Six mutant lines scored the highest yield parameters. Further assessment inferred an inverse relationship between oil and protein content in most of the tested cultivars with high agronomic features. However, four mutant lines recorded high content of oil and protein besides their high seed yield as well, which elect them as potential candidates for large-scale evaluation. The correlation among examined parameters was further confirmed *via* principal component analysis (PCA), which inferred a positive correlation between the number of pods, branches, seeds, and seed weight. Conversely, oil and protein content were inversely correlated in most of yielded mutant lines. Together, those findings introduce novel soybean lines with favorable agronomic traits for the market. In addition, our research sheds light on the

value of using gamma rays treatment in enhancing genetic variability in soybean and improving oil, protein contents and seed yield.

## INTRODUCTION

Soybean (*Glycine max* L. 2n = 2x = 40), a versatile and nutritious legume, holds immense value as a staple in the realms of human food, livestock feed, and oil production (*Singh, Nelson & Chung, 2007*). With its rich composition of essential nutrients, including high-quality protein, healthy fats, and a variety of vitamins and minerals, soybeans have become an indispensable ingredient in countless culinary creations worldwide. Oil and proteins account for 20% and 40%, respectively, of soybean seed contents (*Yazdi-Samadi, Rinne & Seif, 1977*). As a dietary choice for humans, soybeans offer an array of health benefits, promoting heart health, supporting weight management, and contributing to overall well-being. Oil and protein contents in soy cover 29% of vegetable oil consumption and 70% of world protein meal (*Erickson, 2015*). Simultaneously, soybeans play a vital role as a vital protein source in livestock feed, enhancing animal growth and productivity. Soybean also has pharmaceutical and industrial advantages (*Chaudhary et al., 2015*). Previous reports indicated that cultivated soybean lines have an oil content range of 8 to 28%, while protein content ranges from 34% to 57% of the total seed dry mass (*Boerma & Specht, 2004*). Furthermore, soybean oil, extracted from the seeds, serves as a valuable source of edible oil, widely used in cooking and food processing, while also finding applications in industrial sectors. The value of soybeans, therefore, extends far beyond their humble appearance, making them an integral part of our global food and agricultural systems.

Mutation breeding is one of the effective techniques used for introducing and improving desired traits such as yield quality and quantity in economic crops (*Mohan Jain & Suprasanna, 2011*; *Raina et al., 2017*; *Raina, Khursheed & Khan, 2018*; *Amin et al., 2019*). By deliberately inducing specific mutations in plant genomes, scientists can accelerate the natural process of evolution and introduce desirable traits into crops. These directed mutations enable the development of plants with improved yield, disease resistance, tolerance to abiotic stresses such as drought or heat, and enhanced nutritional value. They also facilitate the reduction of chemical inputs, such as pesticides or fertilizers, contributing to sustainable and environmental-friendly farming practices (*Mohan Jain & Suprasanna, 2011*). Directed mutations provide a targeted and efficient approach to crop breeding, allowing researchers to tailor plant characteristics to meet the ever-growing demands of a changing world and ensure food security for a rapidly expanding global population.

The use of gamma rays in plant breeding has proven to be a valuable tool for improving plants and enhancing agricultural productivity (*Ghareeb et al., 2022*). Gamma rays, a form of ionizing radiation, can induce random mutations in plant DNA, leading to genetic variations that can result in desirable traits. By subjecting plants to controlled doses of

gamma rays, we can generate a wide range of mutations, some of which may confer beneficial characteristics such as increased yield, disease resistance, or improved tolerance to environmental stresses. The advantage of gamma ray mutagenesis lies in its ability to induce a high frequency of mutations throughout the entire genome, providing a broad spectrum of genetic variability for selection. Although the process is random, careful screening and selection of mutated plants allow breeders to identify and propagate those with the desired traits (*Ghareeb et al., 2022*). Gamma ray-induced mutagenesis offers an effective and cost-efficient approach to plant breeding, contributing to the development of new crop varieties that are better suited to meet the challenges of a changing climate and increasing food demands.

Breeding soybean with high oil and protein is favored by growers and consumers as well. Due to the negative relationship between protein and oil content, simultaneous breeding for both traits using conventional breeding has been challenging (*Hwang et al., 2014*). Conventional breeding in soybean has also been a challenge due to botanical features such as small fragile flowers and condensed canopy, which makes it tedious to perform the process of hybridization. Improving soybean traits have been successfully achieved using mutational breeding (*Wilcox et al., 2000*; *Mudasir Hafiz & Tyagi, 2010*; *Malek et al., 2012*). Such breeding technology saves time and ensures results compared to the traditional methods of breeding (*Bolbhat & Dhumal, 2009*).

Increased seed yield, oil and protein content are among the ultimate goals of soybean breeders. Understanding the relationship between characters is crucial for selecting the genotypes favored for yield improvement programs (*Kumar & Shukla, 2002*). The present study aimed to use gamma irradiation with different doses to induce useful mutations and trial to break the negative relationship between protein and oil content for the development of new varieties possessing high protein and oil content. The principal component analysis has been employed to define the positive/negative correlations among soybean mutant lines based on their variable traits that are used for selection. As a traditional method, clustering analysis is employed for grouping genotypes into various groups based on distinct features which facilitate the direct selection of parents for a specific breeding program. PCA has indicated the presence of two mutant lines with elevated oil and protein content values compared to their parental ancestors. Several other lines were selected based on their ultimate levels of oil and protein content.

# MATERIALS AND METHODS

## Plant materials

This research was conducted at El-Gemmeiza, Agriculture Research Station, ARC, Egypt during the three consecutive summer seasons of 2017, 2018 and 2019. High-yielding soybean commercial varieties (Giza 21, Giza 22, Giza 82, Giza 83) alongside promising advanced line (Line 117) were used in this study. The utilized genotypes were obtained from the Legume Crops Research Department, Institute of Field Crops, ARC, Giza, Egypt (Table 1).
**Table 1  The pedigree and origin of studied cultivars.**

| Genotype | Pedigree | Maturity group | Flower color | Origin |
|----------|----------|----------------|--------------|--------|
| Giza 21 | Crawford X Celest | IV (120 days) | Purple | FCRI |
| Giza 82 | CrawfordXMppel Presto | III(105 days) | Purple | FCRI |
| Giza 22 | Forest X Crawford | IV (120 days) | Purple | FCRI |
| Giza 83 | Selection from MBB 133 | III(105 days) | White | FCRI |
| Line 117 | D89-8940 X Giza 111 | III (105 days) | White | FCRI |

**Notes.**
FCRI,  Field Crops Research Institute.

## Physical mutagen treatment

The seeds of the soybean cultivars were treated with the following doses of gamma rays 50, 100, 200 and 300 Gy with the radioisotope $Co^{60}$ source (Gamma chamber Model-900 supplied by Nuclear Research Center, Inshas, Egypt). These doses were selected based on the germination test conducted prior to the current study in which we considered the 50 Gy and the 300 Gy as the minimum and the maximum doses with least inhibition/most germination. The dose was applied at the rate of 5.6 Gy/minute.

## Field trial and data collection

The irradiated seeds of soybean cultivars (200 seeds for each cultivar) were grown independently in rows at the end of May of each growing season. Non-treated seeds were also grown at individual lines to serve as a control treatment. The assessed genotypes were sown in three replicates using Randomized Complete Block Design (RCBD). Each genotype was sown in 6-m long row with 0.65-m wide and 0.2-m between hills. Each hill was sown with four seeds and thinned to two seedlings after full emergence. Phosphorus and potassium fertilizer were added during soil preparation at a rate of 100 kg $P_2O_5$ $ha^{-1}$ as superphosphate (15.5% $P_2O_5$) and 100 kg $K_2O$ $ha^{-1}$ as potassium sulfate (48% $K_2O$). Nitrogen fertilizer was applied in the form of ammonium sulfate (20.5 N%) at a rate of 70 kg N $ha^{-1}$ in two equal doses before the first and second irrigations. The seeds were inoculated with the microbial symbiont *Rhizobium japonicum* immediately before sowing. The field trials were irrigated using surface irrigation with 10 days intervals.

The M2 plants were individually screened for morphological changes for further selection. The selected mutants were sorted out based on the following parameters: dwarf (D), semi-dwarf (SD), high number of pods (HNP), high number of seeds (HNS), heavy seeds (HS), height of first pod (HFP), long stem (LS), small seeds (SS), medium seeds (MS) and seed shape (SP).

The selected M2 mutant lines were sown to obtain the M3 generation. The M3 seeds were grown in three replications with 20 seeds per replicate in a randomized complete block design. The growing distance was 65 cm between rows and 20 cm between seeds. Data were collected at the seed maturity stage. The collected data included the previously described morphological characters besides the oil content and the total protein content in each replicate.

Table 2 The type and number of various mutants produced in M2 and M3 generations of the five soybean varieties using gamma rays.

| Mutant type | Varieties | | | | | | | | | | Total | |
|---|---|---|---|---|---|---|---|---|---|---|---|---|
| | Giza21 | | Giza22 | | Giza82 | | Giza83 | | Line117 | | | |
| | M2 | M3 | M2 | M3 | M2 | M3 | M2 | M3 | M2 | M3 | M2 | M3 |
| Dwarf | 5 | 3 | 2 | 1 | 3 | 2 | 3 | 2 | 4 | 1 | 17 | 9 |
| Semi-dwarf | 1 | 1 | 1 | 1 | – | – | – | – | 4 | 1 | 6 | 3 |
| High number of pods | 4 | 4 | 1 | 1 | 5 | 4 | 2 | 1 | 2 | 1 | 14 | 11 |
| High number of seeds | 3 | 2 | 3 | 3 | 2 | 2 | 6 | 5 | 3 | 3 | 17 | 15 |
| Heavy seeds | 4 | 2 | 3 | 1 | 3 | 2 | 1 | – | 1 | 2 | 12 | 7 |
| Height of the first pod | 1 | 1 | 2 | – | 1 | 1 | 2 | 1 | 1 | – | 7 | 3 |
| Long stem | – | – | 1 | 1 | 1 | 1 | – | – | 2 | – | 4 | 2 |
| Small seeds | – | – | 5 | 5 | 2 | 2 | – | – | – | – | 7 | 7 |
| Medium seeds | – | – | – | – | – | – | – | – | 4 | 2 | 4 | 2 |
| Seed shape | – | – | – | – | – | – | 1 | 1 | – | – | 1 | 1 |
| Total | 18 | 13 | 18 | 13 | 17 | 14 | 15 | 10 | 21 | 10 | 89 | 60 |

## Determination of oil percentage and protein content

To determine the % soybean oil, the soybean seeds were cleaned and ground by Moulinex blender Type 716 (France) to pass through a 1 mm sieve. The powder was defatted by hexane in a Soxhlet extractor (*AOAC, 2012*). The total nitrogen in soybean supernatant was measured by the Kjeldahl method (*Saad et al., 2020*) which is then used to estimate the crude protein content by applying a conversion factor (6.25) to the result.

## Statistical analysis

The collected data were statistically analyzed using the analysis of variance (ANOVA) method, and Tukey's HSD *post hoc* test was further used to define significance among treatments at the probability level 5% ($P < 0.05$). The statistical analyses were performed using R statistical software version 4.1.1.

## RESULTS

### Characterization of M2 plants

The mutagenicity of gamma radiation was tested on plants from M2 populations. Different promising derivative mutants were identified in M2 populations of the mutated five soybean genotypes. These mutants involved dwarf, semi-dwarf, a high number of pods, a high number of seeds, heavy seeds, the height of the first pod, long stem, small seeds, medium seeds and seed shape (Table 2). Screening of M2 populations revealed that the treatments with gamma rays mutagens produced 89 mutants of different types (Tables 2 and 3). Results indicated that the using of high doses of gamma radiation (200 Gy and 300 Gy) produced a high number of mutants than the lower doses (50 Gy and 100 Gy).

### Characterization of M3 mutants

Morphological parameters were used to determine the performance of selected M3 mutants derived from used soybean varieties such as yield characters, oil content and protein content.

**Table 3** The number of $M_2$ and $M_3$ mutants produced by different doses of gamma rays with the five soybean varieties.

| Treatments | Varieties | | | | | | | | | | Total | |
|---|---|---|---|---|---|---|---|---|---|---|---|---|
| | Giza21 | | Giza22 | | Giza82 | | Giza83 | | Line117 | | | |
| | M2 | M3 | M2 | M3 | M2 | M3 | M2 | M3 | M2 | M3 | M2 | M3 |
| 50 Gy | 3 | 2 | 4 | 3 | 3 | 2 | 4 | 2 | 4 | 1 | 18 | 10 |
| 100 Gy | 5 | 3 | 5 | 4 | 5 | 3 | 3 | 2 | 5 | 2 | 23 | 14 |
| 200 Gy | 5 | 4 | 5 | 2 | 4 | 4 | 4 | 3 | 6 | 4 | 24 | 17 |
| 300 Gy | 5 | 4 | 4 | 4 | 5 | 5 | 4 | 3 | 6 | 3 | 24 | 19 |
| Total | 18 | 13 | 18 | 13 | 17 | 15 | 15 | 10 | 21 | 10 | 89 | 60 |

The analysis of variance displayed highly significant differences ($p < 0.001$) among the derived mutants for all evaluated traits at levels ($p < 0.001$), while the differences among the three replicates were nonsignificant ($p > 0.05$) for all traits (Table 4). Moreover, the tested genotypes showed highly significant differences based on Tukey's HSD analysis. The results showed that twenty M3 soybean mutants and their corresponding five parent cultivars showed variable performance based on the different evaluated traits. The highest values for plant height in all mutants and their corresponding parents were recorded in G22-LS-1 and H117-HNS-1 mutants, while the lowest value was recorded with the G83-SD-1 mutant (Fig. 1A). The uppermost values for first pod height in all mutants and their corresponding parent were recorded with G21-HNS-1 and H117-HNP-3 mutants, while the lowest values were assigned for G21-HNP-3 and G82-HNP-1 mutants (Fig. 1B). Interestingly, the results elucidated that the mutant H117-HNS-2 showed the highest number of branches per plant, while its corresponding parent recorded the lowest number of branches per plant (Fig. 1C). The superior number of pods per plant was recorded with H117-HNS-1 and H117-HNS-2 mutants, while Gizza-21 and Gizza-22 parents recorded the lowest number of pods/plant as presented in Fig. 2A. Economically, the H117-HNS-2 mutant possessed the highest number of seeds per plant as shown in (Fig. 2B) and the highest 100-seed weight in (Fig. 2C). The highest seed weight per plant was recorded with the H117-HNS-2 mutant (Fig. 3A). The G82-D-1 and H117-HNS-1 mutants possessed the highest oil content (Fig. 3B), while the H117-HNP-1 and H117-HNS-1 mutants recorded the highest protein content (Fig. 3C).

The reduction or increase of the studied traits expressed as a percentage compared with the parental cultivars for M3 soybean mutants waspresented in (Table 5). The results depictedthat the maximum increase in plant height (60.85 cm) was recorded with the H117-HNS-1 mutant obtained by 300 Gy in Line 117. On the other hand, the greatest reduction in plant height ($-27.72$ cm) was recorded with the G83-SD-1 mutant obtained by 200 Gy in Giza 83. The highest increase in first pod height (81.08 cm) was recorded with the G21-HNS-1 mutant obtained by 100 Gy in Giza 21. While the highest reduction in first pod height ($-25.49$ cm) was recorded with the G22-HNS-3 mutant obtained by 50 Gy in Giza 22. The highest increment in the number of branches/plant (250) was recorded with the H117-HNS-2 mutant obtained by 100 Gy in Line 117. While the greatest reduction in the number of branches per plant ($-16.67$) was recorded with the G21-HNS-1

**Table 4  Analysis of variance (presented as mean squares (MS) and probability (Sig.)) of the evaluated traits for twenty M3 mutants and their corresponding parents.**

| Sources of variation | df | Plant height | | Hight of first pod | | Numer of branches per plant | | Number of pods per plant | | Number of seeds per plant | |
|---|---|---|---|---|---|---|---|---|---|---|---|
| | | M.S. | Sig. | M.S. | Sig. | M.S. | Sig. | M.S. | Sig. | M.S. | Sig. |
| Replications | 2 | 2.093 | 0.830 | 2.773 | 0.516 | 0.493 | 0.448 | 26.76 | 0.287 | 45.70 | 0.427 |
| Mutants | 24 | 620.5 | <0.001 | 33.741 | <0.001 | 2.331 | <0.001 | 2,290 | <0.001 | 13,696 | <0.001 |
| Error | 48 | 11.177 | | 4.134 | | 0.604 | | 20.89 | | 52.700 | |
| Total | 74 | 208.6 | | 13.70 | | 1.161 | | 756.9 | | 4,477 | |

| Sources of variation | df | 100 seed weight | | Seed weight per plant | | Oil percentage | | Protein content | |
|---|---|---|---|---|---|---|---|---|---|
| | | M.S. | Sig. | M.S. | Sig. | M.S. | Sig. | M.S. | Sig. |
| Replications | 2 | 0.169 | 0.869 | 1.664 | 0.569 | 0.066 | 0.167 | 0.216 | 0.066 |
| Mutants | 24 | 11.59 | <0.001 | 277.0 | <0.001 | 20.54 | <0.001 | 51.99 | <0.001 |
| Error | 48 | 1.204 | | 2.916 | | 0.035 | | 0.075 | |
| Total | 74 | 4.546 | | 91.78 | | 6.685 | | 19.82 | |

mutant obtained by 100 Gy in Giza 21. The highest increment in the number of pods/plant (258) was recorded with the H117-HNS-1 mutant obtained by 300 Gy in line 117 while the highest reduction in the number of branches/plant (−16.67) was recorded in the G21-HNS-1 mutant obtained by 100 Gy in Giza 21. The highest increment in the number of seeds/plant (282.4) was recorded with the G83-HNS-1 mutant obtained by 200 Gy in Giza 83 while the highest reduction in the number of seeds/plant (0.77) was recorded with the G22-LS-1 mutant obtained by 100 Gy in Giza 22. The highest increment in the 100 seed weight (68.75) was recorded with the H117-HNS-2 mutant obtained by 100 Gy in Line 117 while the highest reduction in the 100 seed weight (−18.35) was recorded with the G82-HNS-1 mutant obtained by 300 Gy in Giza 82. The highest increase in the seed weight/plant (550) was recorded with the H117-HNS-2 mutant obtained by 100 Gy in Line 117 while the highest reduction in the seed weight/plant (−9.2) was recorded with the G22-LS-1 mutant obtained by 100 Gy in Giza 22. The highest increase in the oil percentage (51.16) was recorded with the H117-HNS-1 mutant obtained by 300 Gy in Line 117 while the highest reduction in the oil percentage (−4.2) was recorded with H117-HNS-3 mutant obtained by 300 Gy in Line 117. The highest increase in the protein content (38.02) was recorded with the H117-HNP-1 mutant obtained by 200 Gy in Line 117 while the highest reduction in the protein content (−0.07) was recorded with G82-D-1 mutant obtained by 100 Gy in Giza 82.

## Clustering analysis

The clustering analysis grouped the parents and the emerged mutants into four groups based on seed yield and contributed traits (Fig. 4). The first group (A) included the H117-HNS-2 mutant which displayed the highest values in most agronomic traits. The second group (B) included G83-HNS-1, G82-HNP-1, G82-HNS-1, G21-HNP-3 and G21-HNP-3 which exhibited high agronomic performance. The third group (C) comprised G83-HNP-1,

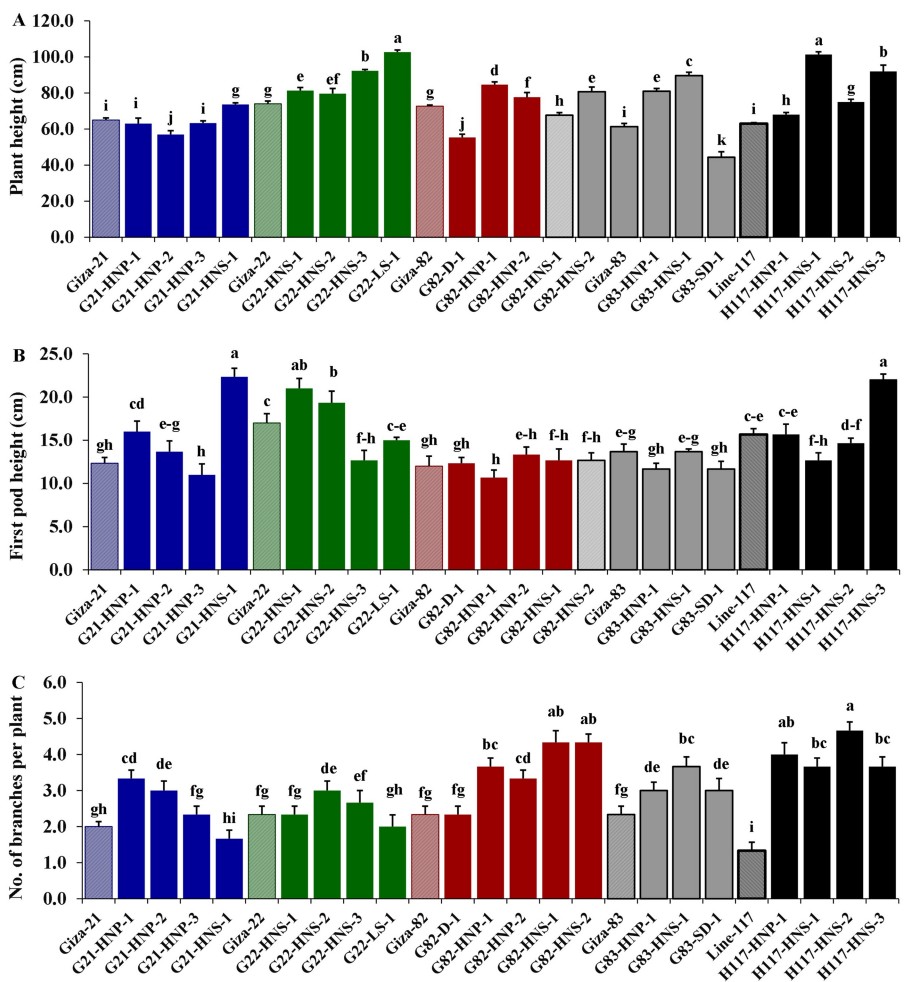

**Figure 1 Performance of twenty M3 soybean mutants and their corresponding parental cultivars for plant height (A), first pod height (B) and number of branches/plant (C).** Different letters on the column differ significantly by Tukey's HSD ($P < 0.05$) and the bars on the top of the columns correspond to SE.

H117-HNS-3, G21-HNP-1, G21-HNP-2, G22-HNS-3, G83-SD-1, G22-HNS-1, G22-HNS-2, G82-D-1, H117-HNP-1, G82-HNP-2 and G82-HNS-2 mutants with intermediate agronomic performance. The fourth group contained Giza 21, Giza 83, Line 117, Giza 22, Giza 82, G21-HNS-1 and G22-LS-1 genotypes with the lowest agronomic performance. On the other hand, the clustering analysis grouped the parents and the emerged mutants into five groups based on oil content (Fig. 5). The first group (A) included G82-D-1 and H117-HNS-1 with the highest oil content. The second group (B) contained G21-HNP-3, G22-HNS-2, G82-HNP-1, G82-HNP-2, and G82-HNS-1 genotypes with high oil content. The third group (C) included G82-HNS-2, H117-HNS-2, G83-SD-1, and G22-HNS-1 with intermediate oil content. The fourth group (D) included Giza 21, Giza 22, Giza 82, Giza 83, G83-HNS-1, G21-HNS-1, G21-HNP-2, G83-HNP-1, G22-LS-1, G21-HNP-1,

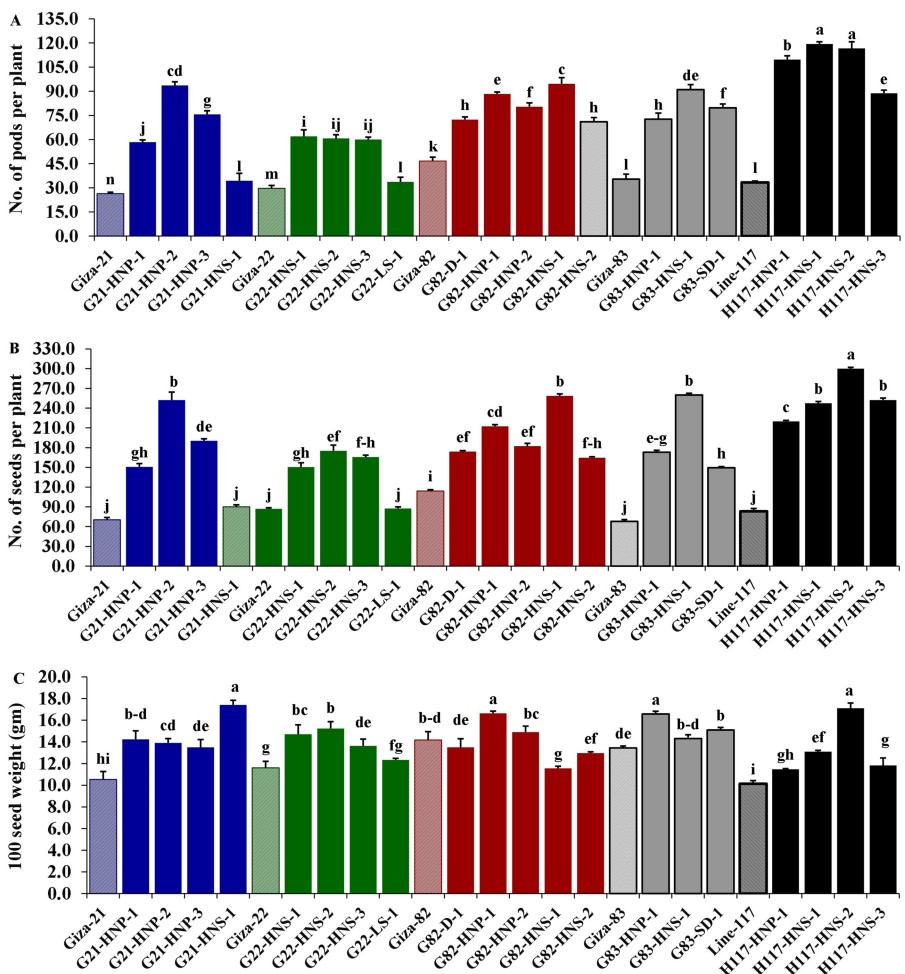

**Figure 2 Performance of twenty M3 soybean mutants and their corresponding parental cultivars for number of pods per plant (A), number of seeds/plant (B) and 100 seed weight (C).** Different letters on the column differ significantly by Tukey's HSD ($P < 0.05$) and the bars on the top of the columns correspond to SE.

and G22-HNS-3 genotypes. The fifth group (E) contained H117-HNS-3, line 117 and H117-HNP-1 with the lowest oil content.

Additionally, the clustering analysis grouped the parents and the emerged mutants into five groups based on protein content (Fig. 6). The first group (A) included H117-HNS-1, H117-HNP-1 and G21-HNS-1 with the highest protein content. The second group (B) comprised H117-HNS-2, G83-HNS-1, G21-HNP-2, G22-LS-1, G83-HNP-1 and H117-HNS-3 genotypes with high protein content. The third group (C) included G22-HNS-3, G21-HNP-1 and G82-HNS-2 mutants with intermediate protein content. The fourth group (D) had G21-HNP-3, G83-SD-1, G82-HNP-2, Giza 21, Giza 83, line 117 and G82-HNS-1 genotypes. The fifth group (E) included the Giza 82, G82-D-1, G82-HNS-2, Giza 22, G22-HNS-1 and G83-HNP-1 genotypes.

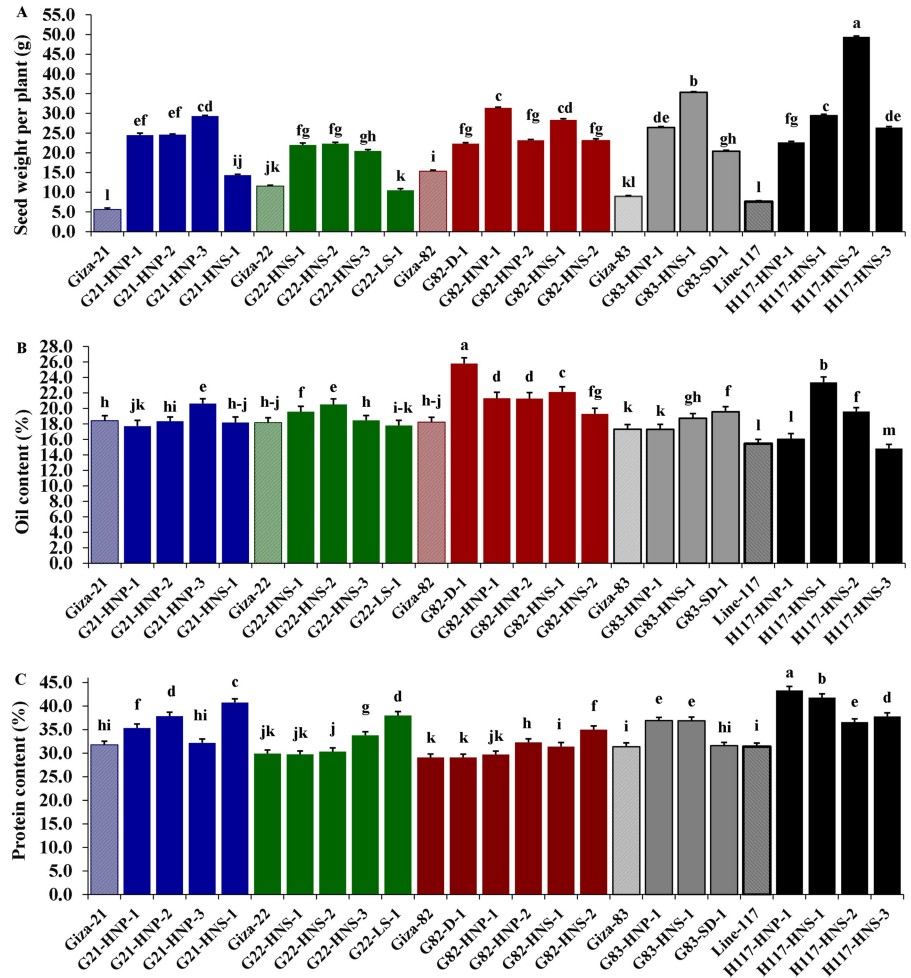

**Figure 3 Performance of twenty M3 soybean mutants and their corresponding parental cultivars for seeds weight per plant (A), oil content (B), and protein content (C).** Different letters on the column differ significantly by Tukey's HSD ($P < 0.05$) and the bars on the top of the columns correspond to SE.

## Interrelationship among evaluated genotypes and studied traits

Principal component analysis (PCA) was conducted to explain the relationship among different traits of the soybean M3 mutants and their corresponding parents. The PC1 and PC2 explored the most variance presenting 80.99% (61.27%% and 19.72% by PC1 and PC2, in the same order), and were used to construct the PC-biplot (Fig. 7). The PC1 exhibited higher variation and seemed to be associated with the evaluated genotypes. PC1 divided the genotypes into two groups on the positive and negative sides of PC1. Most of the studied yield and quality traits were correlated with the genotypes on the positive side of PC1. This indicates the genotypes located on the positive side of PC1 displayed high performance of yield traits as well as oil and protein contents (H117-HNS-2, H117-HNS-1, G83-HNS-1, G82-HNP-1, G83-HNP-1, G82-HNS-2, G21-HNP-2, G21-HNP-1, H117-HNS-3, H117-HNP-1, G21-HNP-3, G82-HNS-1and G82-HNP-2). On the contrary, the remaining genotypes are on the opposite side of PC1 presenting lower performance.

**Table 5 Increment or reduction of the studied traits expressed as percentage compared to the used parental cultivars for M3 soybean mutants.**

| Genotype | PH | PodH | No.BP | No.PP | No.SP | HSW | SWP | Oil | Protein |
|---|---|---|---|---|---|---|---|---|---|
| **Mutants** | | | | Increment or reduction percentage | | | | | |
| G21-HNP-1 | −3.08 | 29.73 | 66.67 | 121.5 | 114.2 | 35.13 | 334.9 | −3.89 | 11.20 |
| G21-HNP-2 | −12.31 | 10.81 | 50.00 | 255.7 | 258.77 | 31.96 | 337.3 | −0.29 | 19.23 |
| G21-HNP-3 | −2.56 | −10.81 | 16.67 | 187.3 | 170.62 | 28.16 | 420.1 | 12.00 | 1.39 |
| G21-HNS-1 | 13.33 | 81.08 | −16.67 | 30.4 | 27.96 | 65.19 | 154.4 | −1.19 | 28.31 |
| G22-HNS-1 | 9.91 | 23.53 | 0.00 | 109.0 | 72.80 | 26.72 | 90.2 | 7.82 | −0.63 |
| G22-HNS-2 | 7.66 | 13.73 | 28.57 | 104.5 | 101.53 | 31.32 | 93.1 | 12.99 | 1.28 |
| G22-HNS-3 | 24.77 | −25.49 | 14.29 | 102.2 | 90.42 | 17.53 | 77.2 | 1.63 | 12.83 |
| G22-LS-1 | 38.74 | −11.76 | −14.29 | 13.5 | 0.77 | 6.32 | −9.2 | −2.04 | 26.91 |
| G82-D-1 | −23.85 | 2.78 | 0.00 | 55.0 | 52.63 | −4.71 | 45.7 | 41.52 | −0.07 |
| G82-HNP-1 | 16.51 | −11.11 | 57.14 | 89.3 | 86.26 | 17.41 | 104.8 | 16.90 | 2.05 |
| G82-HNP-2 | 6.88 | 11.11 | 42.86 | 72.1 | 59.94 | 5.18 | 51.3 | 16.73 | 10.89 |
| G82-HNS-1 | −6.88 | 5.56 | 85.71 | 102.9 | 126.90 | −1835 | 84.8 | 21.34 | 7.86 |
| G82-HNS-2 | 11.01 | 5.56 | 85.71 | 52.1 | 44.44 | −8.47 | 50.2 | 5.81 | 20.19 |
| G83-HNP-1 | 32.07 | −14.63 | 28.57 | 105.7 | 154.4 | 23.33 | 195.9 | −0.12 | 17.66 |
| G83-HNS-1 | 46.20 | 0.00 | 57.14 | 157.5 | 282.4 | 6.45 | 295.5 | 8.20 | 17.56 |
| G83-SD-1 | −27.72 | −14.63 | 28.57 | 125.5 | 119.6 | 12.41 | 128.4 | 12.93 | 0.73 |
| H117-HNP-1 | 7.94 | 0.00 | 200.0 | 229.0 | 164.0 | 13.16 | 198.2 | 3.82 | 38.02 |
| H117-HNS-1 | 60.85 | −19.15 | 175.0 | 258.0 | 196.8 | 29.28 | 288.2 | 51.16 | 33.24 |
| H117-HNS-2 | 19.05 | −6.38 | 250.0 | 250.0 | 260.0 | 68.75 | 550.0 | 26.78 | 16.63 |
| H117-HNS-3 | 46.03 | 40.43 | 175.0 | 166.0 | 202.8 | 16.78 | 247.8 | −4.20 | 20.46 |
| **Parental cultivar** | | | | Mean performance | | | | | |
| Giza 21 | 65.00 | 12.33 | 2.00 | 26.33 | 70.33 | 10.53 | 5.63 | 18.42 | 31.76 |
| Giza 22 | 74.00 | 17.00 | 2.33 | 29.67 | 87.00 | 11.60 | 11.57 | 18.17 | 29.94 |
| Giza 82 | 72.67 | 12.00 | 2.33 | 46.67 | 114.00 | 14.17 | 15.33 | 18.23 | 29.12 |
| Giza 83 | 61.33 | 13.67 | 2.33 | 35.33 | 68.00 | 13.43 | 8.93 | 17.32 | 31.37 |
| Line 117 | 63.00 | 15.67 | 1.33 | 33.33 | 83.33 | 10.13 | 7.60 | 15.46 | 31.38 |

The vectors of traits form acute angles, implying positive correlations and vice versa. Appreciably, there was a clear positive inter-association among the number of pods, the number of branches, the number of seeds, the weight of 100 seeds and seed weight per plant. However, there was a negative relationship between oil content and protein content.

These findings are valuable for plant breeders to build selection criteria on strong positively associated characters. Likewise, the heatmap and hierarchical clustering based on the yield traits, oil content and protein content divided the evaluated genotypes into distinct clusters (Fig. 8). The genotypes H117-HNP-1, H117-HNS-3, H117-HNS-2, H117-HNS-1, G82-HNS-1, G83-HNS-1, G82-HNP-1, G82-HNP-2 and G82-D-1 possessed the uppermost values for studied traits (presented in red). Conversely, the remaining genotypes possessed low values (blue values). Interestingly, the derivative mutant G82-HNP-1, G82-D-1, G82-HNP-2, and G82-HNS-1 displayed high oil and protein content alongside high seed yield (Fig. 9).
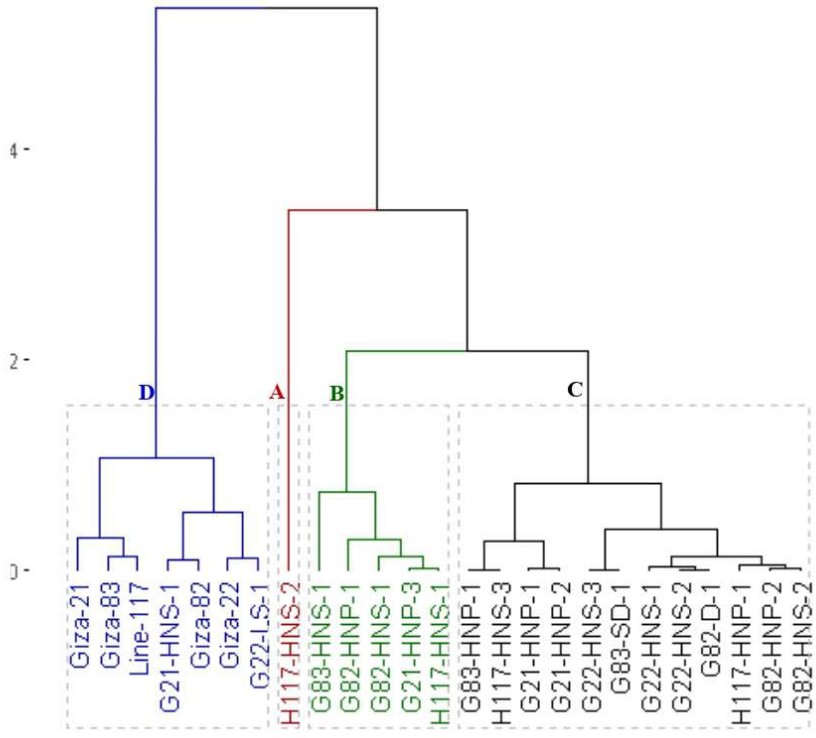

**Figure 4** Dendrogram of the assessed 25 soybean genotypes (twenty M3 mutants and their parental cultivars) based on seed yield and contributed traits.

## DISCUSSION

In soybean breeding, development of new cultivars with improvement characters such as yield, oil and protein contents is considered the main objective. The influences of gamma radiation on certain plants and seed properties have been investigated (*Canbay & Bardakci, 2011*; *Dixit et al., 2011*). In this study, several evaluated traits were positively impacted by gamma rays in the M3 soybean-generated mutants. The reduction or increase of the plant height trait expressed as the percentage compared with their parental cultivars for M3 soybean mutants resulting from the use of different doses of gamma rays ranged between −23.85 to 60.85, respectively. *Khan & Tyagi (2013)* reported that the treatment of soybean seeds with gamma rays led to the majority of the mutants in M2–M4 being taller than the control plant. *Fahmy et al. (1997)* deduced that plant height was inversely associated with increasing gamma-ray exposures. *Khan & Tyagi (2013)* indicated that there was a slight improvement in plant height following the application of three doses of gamma radiation (10, 20 and 30 kR) to three varieties of soybean.

The exposure to gamma radiation can be used to induce mutations and thereby generate genetic variation from which desired mutants may be selected (*Khan et al., 2005*; *Hanafiah et al., 2011*; *Badr et al., 2014*; *Dhakshanamoorthy, Selvaraj & Chidambaram, 2015*; *Amri-Tiliouine et al., 2018*; *Ghareeb et al., 2022*; *Zafar et al., 2022*). Mutation breeding has been

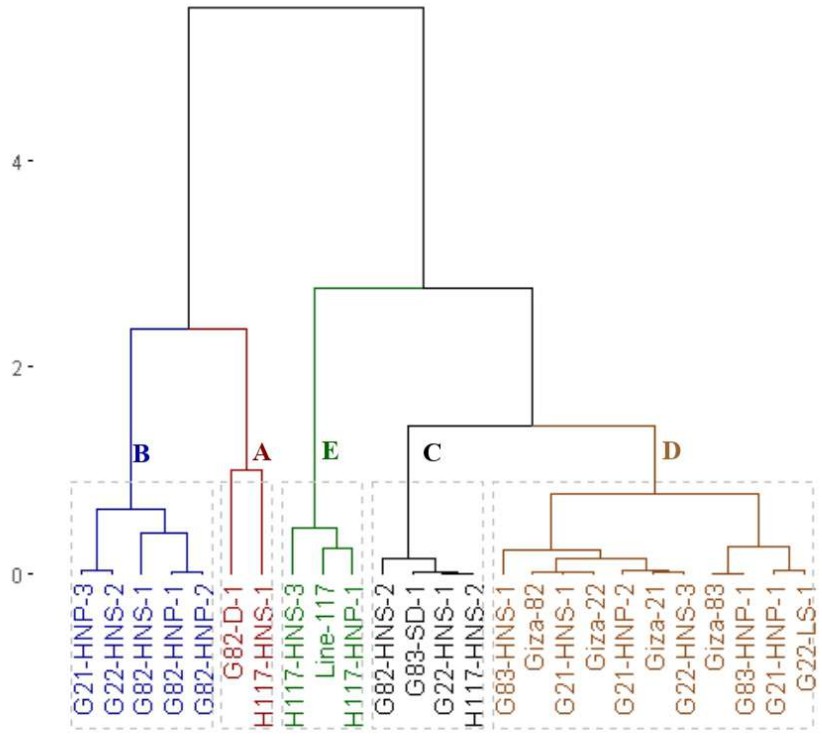

**Figure 5 Dendrogram of the assessed 25 soybean genotypes (twenty M3 mutants and their parental cultivars) based on oil content.**

used to develop a wide variety of improved crop varieties, including wheat, rice, corn, soybeans, potatoes, and tomatoes. These varieties have helped to increase food production and improve food security around the world. Some of the beneficial changes that can be brought about by mutation breeding include: Increased yield (*Ahloowalia, Maluszynski & Nichterlein, 2004*; *Abaza et al., 2020*; *Pandit et al., 2021*), Improved resistance to pests and diseases (*Kozjak & Meglič, 2012*; *Raina & Danish, 2018*), better nutritional value (*Raina & Danish, 2018*; *Sarsu, 2020*) and tolerance to abiotic stresses (*Robles, Micol & Quesada, 2015*). In soybean, the following parameters (number of pods per plant, seed weight, and seeds per pod) are important for genotype selection. The number of pods/plant increased in the H117-HNS-1 mutant (258) by using the gamma rays 300 Gy while the lower dose (100 Gy) led to a decline in the number of branches per plant (−16.67) with G21-HNS-1 mutant compared the parent genotypes. *Zakri & Jalani (1988)* used gamma rays treatment with two soybean cultivars and reported that the number of pods/plant in mutant P630- 2 increased up to 390% of the control. Interestingly, the treatment of two different genotypes (line 117 and Giza 22) with the same dose of gamma rays (100 Gy) increased the seed weight/plant (550) with the H117-HNS-2 mutant and decreased the seed weight/plant (−9.2) with G22-LS-1 mutant. This indicates that the effect of irradiation treatment may vary according to the genotype.
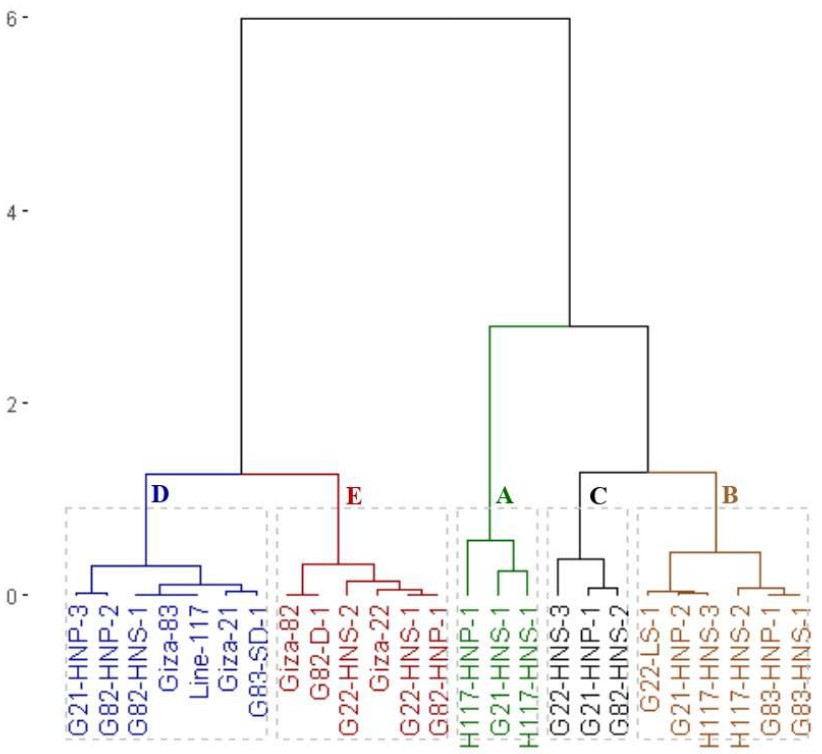

**Figure 6** Dendrogram of the assessed 25 soybean genotypes (twenty M3 mutants and their parental cultivars) based on protein content.

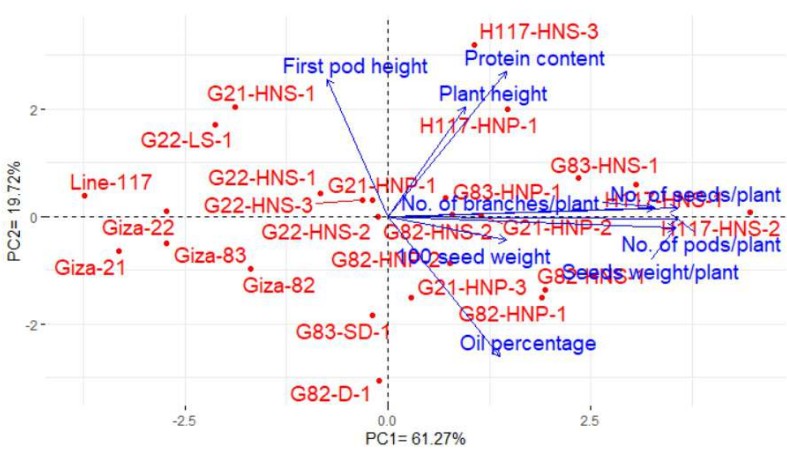

**Figure 7** PC-biplot for the studied traits of the evaluated twenty M3 mutants and their corresponding parent cultivars.

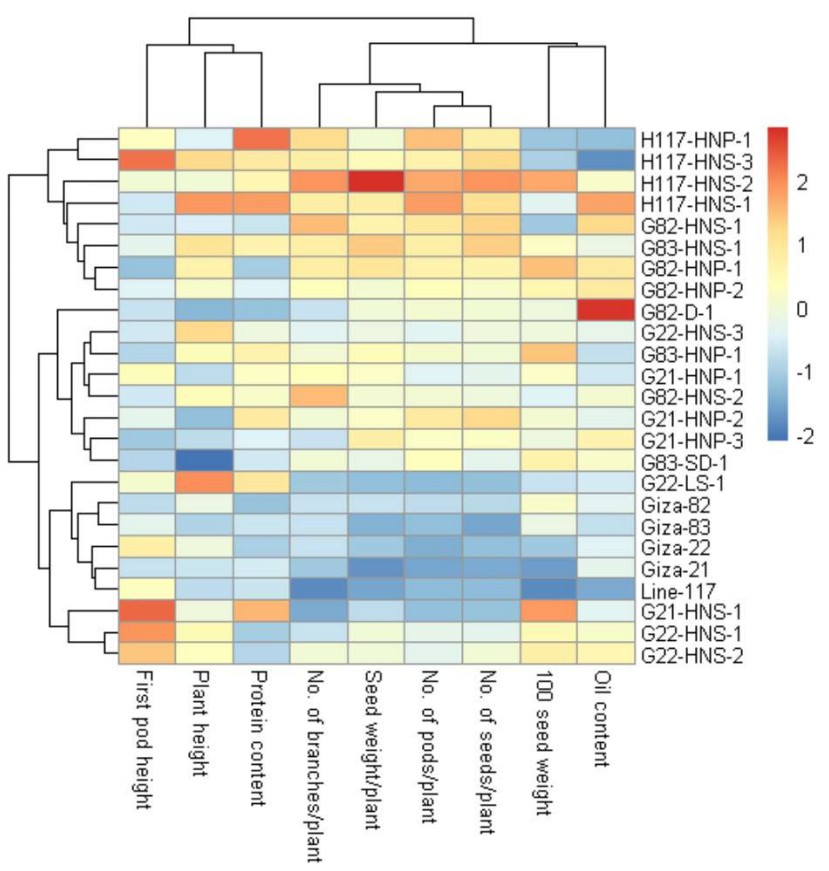

**Figure 8** **Heatmap and hierarchical clustering divide the evaluated twenty M3 mutants and their corresponding parent cultivars into different clusters based on the evaluated traits.** Red and blue colors imply high and low values for related traits, respectively.

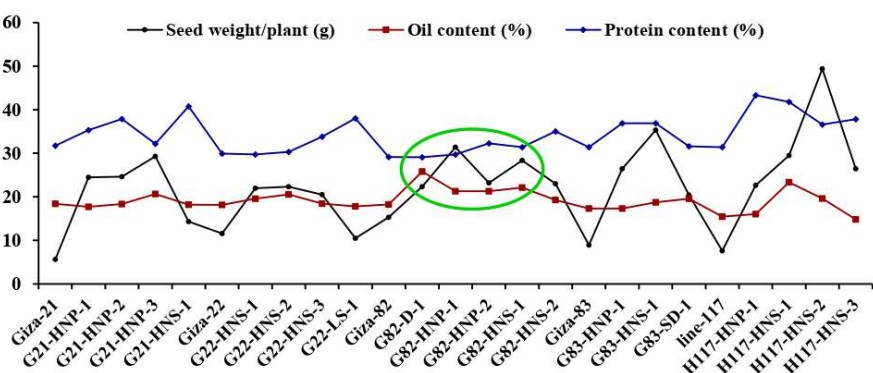

**Figure 9** **Mean performance of twenty M3 soybean mutants and their corresponding parent cultivars for oil content and protein content.**

According to *Harb (1990)*, the number of seeds produced per plant did not increase on average as the dose was raised. However, doses 4 and 8 kR produced more seeds per plant than parental plants in the cultivar Adams. Likewise, *Rajput (1987)* found that the gamma rays had a depressive effect on the number of seeds per plant. *Krausse (1989)* developed multiple mutants and demonstrated that Dorado was the superior derived mutant with greater yields compared to the parental cultivar. *Kumar & Lal (2001)* found that the highest magnitude of variation in grain yield was induced by 30 kR gamma rays. The same gamma rays dose (300 Gy) with line 117 produced two extremely different mutants. The first H117-HNS-1 mutant possessed the highest increase in the oil percentage (51.16) and the second H117-HNS-3 mutant possessed the highest reduction in the oil percentage (−4.2).

Protein content trait increased in the H117-HNP-1 mutant using a 200 Gy gamma irradiation dose. The highest reduction in the protein content (−0.07) was achieved in the G82-D-1 mutant by the use of 100 Gy with the Giza 82 genotype. Previous studies that used gamma irradiation with several plant species reported an increase in oil content which support our results (*Sattar et al., 1989*). Conversely, *Dixit et al. (2011)* found that the gamma radiation had no effects on the oil and protein of soybean seeds. Some reports indicated that high doses of gamma rays are frequently inhibitory (*Mohajer et al., 2014*), whereas low doses are occasionally stimulatory (*Vasconcelos et al., 2006*).

Clustering analysis based on yield, oil content, and protein content traits grouped the 25 soybean genotypes (twenty M3 mutants and their corresponding parental cultivars) into four, five and five different clusters, respectively. This indicates that the 25 soybean genotypes displayed a wide range of diversity in terms of yield, oil content, and protein content traits. This clustering will facilitate the selection of parents for a specific trait in hybridization programs. Several studies reported the use of clustering analysis to group the soybean genotypes into different clusters (*Cui et al., 2001*; *Iqbal et al., 2008*; *Ojo, Ajayi & Oduwaye, 2012*). The dendrograms tend to group the mutants with similar features together. Similar findings were also presented in soybean and other crops (*Iqbal et al., 2008*; *Abdullah et al., 2011*; *Latif et al., 2011*; *ElShamey et al., 2022*; *Essa et al., 2023*). The current results demonstrated that induced mutations play a valuable role in developing genetic variants in crop plants. PCA-biplot and hierarchical clustering are effective statistical tools to understand the relationship betweenevaluated genotypes and studied traits (*El-Sanatawy et al., 2021*; *Mansour et al., 2021*; *Kamara et al., 2022*; *Sakran et al., 2022*; *Al-Khayriet al., 2023*). The analyses displayed the superiority of derivative mutantsH117-HNS-2, H117-HNS-1, G83-HNS-1, G82-HNP-1, G83-HNP-1, G82-HNS-2, G21-HNP-2, G21-HNP-1, H117-HNS-3, H117-HNP-1, G21-HNP-3, G82-HNS-1, and G82-HNP-2. Interestingly, the mutants G82-HNP-1, G82-D-1, G82-HNP-2, and G82-HNS-1 recorded high oil percentage and protein contents alongside high seed yield which is infrequent in soybean genotypes.

This investigation will support conventional breeding experiments (*Hassanin et al., 2020*) to improve the soybean genotypes, more recent genetic techniques, such as genome editing techniques (*Abdelnour et al., 2021*; *Raza et al., 2022*), molecular markers (*Eldomiaty & Mahgoub, 2021*; *Al-Khayri et al., 2022*; *Essa et al., 2023*) and phylogenetic analysis (*Fang et al., 2010*; *Hassanin et al., 2022*) are essential to investigate the genetic variation among
several plant species, which could efficient in the genetic improvement of soybean genotypes for different desirable traits.

## CONCLUSION

The current investigation reported high levels of phenotypic variations for the investigated traits among the twenty soybean mutants and their corresponding parents. These mutant lines could be candidates for further genetic improvement of soybean genotypes in breeding programs. Among the studied traits, a positive correlation was observed between protein content and plant height, and between the 100 seeds' weight and oil content. Therefore, effective phenotypic selection could be applied to the genetic improvement of soybean by considering these favorable attributes together. The majority of the studied traits revealed positive associations with each other, which will allow the combined improvement of these traits by selecting only easily measurable and highly heritable characters. Cluster analysis using investigated traits grouped the 20 soybean mutants and five parent genotypes into four, five and five different clusters based on yield and contributed traits, oil content and protein content, respectively.

### Funding

This work was supported by the Princess Nourah bint Abdulrahman University Researchers Supporting Project number (PNURSP2023R318), Princess Nourah bint Abdulrahman University, Riyadh, Saudi Arabia and by the Deanship of Scientific Research, Vice Presidency for Graduate Studies and Scientific Research, King Faisal University, Saudi Arabia (Project number: GRANT 3,203). The funders had no role in study design, data collection and analysis, decision to publish, or preparation of the manuscript.

### Grant Disclosures

The following grant information was disclosed by the authors:
The Princess Nourah bint Abdulrahman University Researchers: PNURSP2023R318.
Princess Nourah bint Abdulrahman University, Riyadh, Saudi Arabia.
The Deanship of Scientific Research, Vice Presidency for Graduate Studies and Scientific Research, King Faisal University, Saudi Arabia: GRANT 3,203.

### Competing Interests

Elsayed Mansour and Diaa Abd El-Moneim are Academic Editors for PeerJ.

### Author Contributions

- Geehan Mohsen conceived and designed the experiments, prepared figures and/or tables, and approved the final draft.
- Said S. Soliman conceived and designed the experiments, prepared figures and/or tables, and approved the final draft.

- Elsayed I. Mahgoub performed the experiments, authored or reviewed drafts of the article, and approved the final draft.
- Tarik A. Ismail performed the experiments, prepared figures and/or tables, and approved the final draft.
- Elsayed Mansour performed the experiments, authored or reviewed drafts of the article, and approved the final draft.
- Khairiah M. Alwutayd analyzed the data, prepared figures and/or tables, and approved the final draft.
- Fatmah A. Safhi analyzed the data, authored or reviewed drafts of the article, and approved the final draft.
- Diaa Abd El-Moneim conceived and designed the experiments, authored or reviewed drafts of the article, and approved the final draft.
- Rahma Alshamrani analyzed the data, prepared figures and/or tables, and approved the final draft.
- Osama O. Atallah analyzed the data, authored or reviewed drafts of the article, and approved the final draft.
- Wael F. Shehata conceived and designed the experiments, prepared figures and/or tables, and approved the final draft.
- Abdallah A. Hassanin conceived and designed the experiments, authored or reviewed drafts of the article, and approved the final draft.

## Data Availability

The raw data is available in the Supplemental Files.

## Supplemental Information

Supplemental information for this article can be found online at http://dx.doi.org/10.7717/peerj.16395#supplemental-information.

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
