# Peer review of "Gamma-rays induced mutations increase soybean oil and protein contents"

_PeerJ, doi:10.7717/peerj.16395_

## Round 0.1 · original submission · Major Revisions

Your manuscript was reviewed by three independent experts in the field. The reviewers found the work interesting but raised several issues which need to be addressed properly in the revision. In addition to this, authors should improve the English language and flow of information of whole manuscript substantially.

Reviewer 1 ·

Basic reporting

Mohsen et al. present an investigation into the use of gamma rays as a mutagen to enhance soybean traits, particularly oil and protein contents. The study provides necessary background information on the importance of soybean and justifies the use of gamma rays for inducing random mutations and generating genetic variations to address agricultural challenges.

Experimental design

The study demonstrates a good experimental design. It spans three summer seasons and includes five soybean genotypes, ensuring a comprehensive evaluation of mutational effects induced by gamma irradiation. The authors describe the experimental layout, spacing, and replication clearly. Established methodologies are used to assess morphological parameters, yield traits, oil content, and protein content, ensuring reliable and comparable data

Table 5. shows interesting information. However, it is not clear what was the general performance of the five parental cultivars. I see their pedigree being mentioned in Table 1. A brief description or citation on why these parental cultivars were chosen or what their performance is like would be very helpful in providing better context for mutants from each line. I see the parental agronomic data in Figures 1-3 but the way it is presented still makes the comparison across the parents harder to comprehend.

Validity of the findings

Line 31 and Line 246, the robust effect may need to be paraphrased or needs to be explained further to claim “robust”. The word robust has a historical meaning in statistics. My understanding is that in this paper, it is being used as an adjective. See- Huber, P. J. (2011). Robust statistics. In International encyclopedia of statistical science (pp. 1248-1251). Springer, Berlin, Heidelberg.

Line 236-240, This is a very broad generalization. For e.g., it looks like oil content and protein content are explained by both PC1 and PC2, which is not discussed in the paper. Also, it is very hard to track the parents and the mutants within the plot. A faceted plot or color-coded plot may communicate this information. And I wonder if authors created a similar plot with all the generation of mutants and not just M3 generation, which can increase the interpretability of the data.

Line 298-299, There are no genetic data shown in this study, so it cannot really indicate the presence of “significant genetic diversity”. This should be rephrased as a speculation and without the word “significant”, as the word significant without a significance test is not suitable.
Line 325, clarify high level of variation is a phenotypic variation

Additional comments

The paper can be drastically improved if flow is improved in discussion section, which at times look a bit dis-organized.
Overall, this research addresses a knowledge gap in soybean improvement and offers a practical approach for augmenting genetic variability. The identification of mutant lines with increased oil and protein contents has significant implications for the soybean industry. The publication of this study in PeerJ is justified.

Reviewer 2 ·

Basic reporting

The manuscript describes the effect of gamma radiation on the agronomic attributes of a number of soybean cultivar, including their protein and oil content. The use of gamma radiation for crop improvement is not new and has demonstrated its potential, yet, the molecular basis of this approach is still not well understood. The approach used in this study is interesting but rather descriptive in nature. Improvements are required in many aspects such as methodology and data interpretation.

Experimental design

The methodology section suffers from a lack of details on a few things that may affect its reproducibility.
1. Section 2.2 Please state the duration of exposure to the radiation.
2. Line 134: the methods for determination of oil and protein content are missing
3. Line 124-125: please provide the information on the watering and fertilising practices.
4. Section 2.6: please indicate clearly if independent experiments were performed in this study (n = ?)

Validity of the findings

The authors have showed that exposure to different duration of gamma radiation have beneficial effects on some of the cultivars/mutants, however, how this was brought about, was not investigated or discussed thoroughly in the manuscript. In fact, a more critical interpretation of the findings of the present study is needed, and one of the ways this can be achieved is by making comparison to previous relevant studies that deals with mutation in crop improvement, and not necessarily limited to soybeans.

1. I believe molecular characterisation of the mutants generated in this study is crucial to confirm their status and stability. Did the authors consider to use molecular markers (such as RFLP) for mutation detection and characterisation?
2. Changes in the protein and oil content (concentration?) were observed but it is unsure if such changes also involve the type and composition of the proteins and oil? (the method of determination of proteins and oil were not included in the manuscript). Such data would elevate the impact of the present study to provide a more comprehensive understanding of the effect of gamma radiation of the agronomic traits and biochemical composition of soybeans.

Additional comments

The experimental approach is interesting but lack of details. The authors are suggested to consider and incorporate the above recommendations in the revised manuscript.

·

Basic reporting

I suggest to change the title as "Gamma-rays induced increase in soybean oil and protein contents.

It is pure mutation breeding experiment. So it is recommended to use the term Mutation Breeding in place of mutational engineering.

Put latest references to support claims in the mutation breeding section. The papers of author Aamir Raina may be helpful in this regard.

Please mention why you chose the minimum dose of 50 Gy and maximum dose of 200 Gy.

Line 298 "five is repeated"
Line 304 and 304 A lot of references are added.

Line 320 Reframe the sentence

How many seeds were irradiated with each dose of gamma rays.

Experimental design

I have already mentioned comments above.

Validity of the findings

It is fine but requires more details on methodology to validate the findings.

Additional comments

None

---

## Round 0.2 · Major Revisions

I evaluated the reviewers' reports and noticed that the authors failed to address the points raised by the reviewers, particularly reviewer 2, and thus the manuscript requires additional revision(s).

Reviewer 1 ·

Basic reporting

The authors have made the changes I suggested in the last review. I recommend its publication in this journal.

Experimental design

no comment

Validity of the findings

no comment

Additional comments

no comment

Reviewer 2 ·

Basic reporting

The authors have responded to the reviewers' comments and made the necessary amendments in the manuscript.

1. There is no clear response to my query on the number of independent experiments (n=?) in lines 156-159 as cited by the author in the rebuttal letter. The number of experiments should be indicated clearly in the figure legend too.

2. The authors were suggested to discuss on how mutation breeding may bring about beneficial changes, in the context of known literature, but only a general statement on the impact of mutation with multiple citations were provided - any interesting findings from the long list of references?

In my opinion, the one-sentence paragraph in the discussion section should have been avoided in the final version of the manuscript.

3. The last sentence in the conclusion section is rather confusing. What exactly is engineering genetic variations - in the context of this study?
"These obtained findings also confirm that not only the genetic background but also induced mutations substantially contribute to engineeringgenetic variations."

Experimental design

No comment

Validity of the findings

No comment

---

## Round 0.3 · Minor Revisions

Reviewer 2 suggested to use the standard nomenclature of replicates therefore the manuscript needs a further revision.

Reviewer 2 ·

Basic reporting

The authors have amended the manuscript based on my previous comments.

Experimental design

For statistical analysis, I am of opinion that standard terms should be used to avoid confusion, i.e., replicates and independent experiments, rather than vague terms like "independent replications" - do the authors refer to replicates within one set of experiment or several experiments were performed at different times?

Validity of the findings

No comment

Additional comments

No comment

---

## Round 0.4 · accepted · Accept

I appreciate authors' effort in revising the manuscript satisfactorily. The current version is suitable for publication in PeerJ.